Objective monitoring of functional recovery after total knee and hip arthroplasty using sensor-derived gait measures

Boekesteijn Ramon r.boekesteijn@maartenskliniek.nl 1 2
Smolders José 3
Busch Vincent 3
Keijsers Noël 1
Geurts Alexander 2
Smulders Katrijn 1
1 Department of Research, Sint Maartenskliniek , Nijmegen , The Netherlands
2 Department of Rehabilitation, Donders Institute for Brain Cognition and Behaviour, Radboud University Medical Center , Nijmegen , The Netherlands
3 Department of Orthopedic Surgery, Sint Maartenskliniek , Nijmegen , The Netherlands
van Dieën Jaap
Electronic publication date: 2022 Sep 28
Publication date: 2022
Volume: 10
Electronic Location ID: e14054
Received 2022 May 2; Accepted 2022 Aug 24
Copyright: ©2022 Boekesteijn et al.
Copyright year: 2022
Copyright holder: Boekesteijn et al.
License: This is an open access article distributed under the terms of the Creative Commons Attribution License, which permits unrestricted use, distribution, reproduction and adaptation in any medium and for any purpose provided that it is properly attributed. For attribution, the original author(s), title, publication source (PeerJ) and either DOI or URL of the article must be cited.
License URL: https://creativecommons.org/licenses/by/4.0/

Keywords: Gait, Osteoarthritis, Arthroplasty, PROMs, Wearable sensors, Accelerometer, Gyroscope

Funding: Innovation Fund of the Sint Maartenskliniek The Innovation Fund of the Sint Maartenskliniek sponsored this study. The funders had no role in study design, data collection and analysis, decision to publish, or preparation of the manuscript.

==============================
Background

Inertial sensors hold the promise to objectively measure functional recovery after total knee (TKA) and hip arthroplasty (THA), but their value in addition to patient-reported outcome measures (PROMs) has yet to be demonstrated. This study investigated recovery of gait after TKA and THA using inertial sensors, and compared results to recovery of self-reported scores of pain and function.

Methods

PROMs and gait parameters were assessed before and at two and fifteen months after TKA (n = 24) and THA (n = 24). Gait parameters were compared with healthy individuals (n = 27) of similar age. Gait data were collected using inertial sensors on the feet, lower back, and trunk. Participants walked for two minutes back and forth over a 6m walkway with 180° turns. PROMs were obtained using the Knee Injury and Osteoarthritis Outcome Scores and Hip Disability and Osteoarthritis Outcome Score.

Results

Gait parameters recovered to the level of healthy controls after both TKA and THA. Early improvements were found in gait-related trunk kinematics, while spatiotemporal gait parameters mainly improved between two and fifteen months after TKA and THA. Compared to the large and early improvements found in of PROMs, these gait parameters showed a different trajectory, with a marked discordance between the outcome of both methods at two months post-operatively.

Conclusion

Sensor-derived gait parameters were responsive to TKA and THA, showing different recovery trajectories for spatiotemporal gait parameters and gait-related trunk kinematics. Fifteen months after TKA and THA, there were no remaining gait differences with respect to healthy controls. Given the discordance in recovery trajectories between gait parameters and PROMs, sensor-derived gait parameters seem to carry relevant information for evaluation of physical function that is not captured by self-reported scores.

Introduction

Walking is essential for many activities of daily living, and a good walking capacity is key for participation in society. Previous reports have identified walking speed as ‘sixth vital sign’, given its correlation with essential health parameters, including quality of life (Schmid et al., 2007), risk of future hospitalization (Montero-Odasso et al., 2005), and mortality (Hardy et al., 2007). In individuals with end-stage osteoarthritis (OA) of the knee and hip, walking capacity is reduced (Thomas, Pagura & Kennedy, 2003), thereby leading to decreased physical functioning and a lower quality of life (Neogi, 2013). As final step in the treatment of severe knee and hip OA, total joint arthroplasty can be performed in order to resolve OA-related symptoms (e.g., pain, stiffness, instability) and improve physical functioning.

Although total knee arthroplasty (TKA) and total hip arthroplasty (THA) are very successful and cost-effective procedures (Ethgen et al., 2004), a subset of patients is dissatisfied with treatment outcome (Gunaratne et al., 2017; Anakwe, Jenkins & Moran, 2011; Nilsdotter, Toksvig-Larsen & Roos, 2009). In addition to patients with identified complications, this includes patients who had an uneventful procedure, but did not achieve their expected level of functional recovery (Gunaratne et al., 2017). Early identification of individuals at-risk of limited functional recovery is crucial in order to enable clinicians to intervene timely, and may help to readjust patient expectations (Tolk et al., 2021). However, it has been challenging to identify these patients. In part, this is due to a lack of outcomes of physical functioning with good psychometric properties (Hossain et al., 2015). Current diagnostics (e.g., radiographs, physical exam, self-reported outcomes) are limited to static or non-weightbearing situations, or are not necessarily reflective of someone’s actual performance during daily life activities (Bolink et al., 2016; Fransen et al., 2019). Moreover, patient-reported outcomes (PROMs) are inherently subjective, largely influenced by pain, and suffer from early ceiling effects (Stevens-Lapsley, Schenkman & Dayton, 2011). Although PROMs often contain subscales related to limitations in activities of daily life, such as KOOS/HOOS-ADL or WOMAC function score, these outcomes seem to be more reliant on a patients’ own reflections on their capacity rather than their actual performance (Fransen et al., 2019). Hence, there is a need for objective data that can bridge this gap in clinical assessment.

As an alternative to these subjective scores, performance-based tests have been proposed to objectively capture physical function. For example, evaluation of sit-to-stand transfers, walking short distances, and stair negotiation has been endorsed by the OARSI as core-activities for individuals with knee and hip OA (Dobson et al., 2013). While these tests are well-suited to quickly obtain a global picture of a patient’s physical function, they are limited to a single outcome measure, being the time to perform the task or activity, completed distance, or number of repetitions. These tests provide no information about compensations or underlying biomechanics relevant to the performance, and thus may lack important details. Wearable, inertial sensors, are promising tools to instrument performance-based tests in order to obtain more detailed insights into physical functioning. These inertial sensors are easy to use, have been proven to be valid and reliable (Kobsar et al., 2020a), do not require lengthy procedures or specialized laboratories, and can be used in clinal settings or even remotely in the home environment (Fransen et al., 2021). Not surprisingly, inertial sensors have gained interest over the past few years to objectively monitor changes in physical function after total knee and hip arthroplasty (Small et al., 2019; Kobsar et al., 2020b). In particular, the focus has been on studying gait recovery (Small et al., 2019; Kobsar et al., 2020b), potentially due to the fact that gait parameters are predictive of limitations in other activities of daily living (Potter, Evans & Duncan, 1995) and gait improvements are an important goal for patients after TKA and THA (Scott et al., 2012). In the same settings, turning could also be evaluated (Boekesteijn et al., 2021), which has been suggested to be even more sensitive to sensorimotor impairments than straight ahead gait (Mancini et al., 2016). However, before such technologies can be clinically adopted, it is important that the derived outcome measures fulfill the following requirements: they must (1) be sensitive to pre-operative impairment, (2) be responsive to interventions aimed at improving mobility, and (3) provide clinically relevant information about physical functioning.

Multiple gait and turning parameters derived from inertial sensors have shown to be sensitive to mobility impairment in end-stage knee and hip OA (Boekesteijn et al., 2021). The next step herein is to evaluate responsiveness of these parameters to unilateral TKA and THA, and to assess whether post-operative function recovers to the level of healthy individuals. While recovery of gait has previously been investigated using inertial sensors at different timepoints after TKA (Fransen et al., 2019; Fransen et al., 2021; Bolink, Grimm & Heyligers, 2015; Senden et al., 2011; Jolles et al., 2012; Kluge et al., 2018; Youn et al., 2020) and THA (Bolink et al., 2016; Reininga et al., 2013; Nelms et al., 2020; Wada et al., 2019), a comprehensive study is lacking that maps the recovery trajectory —including turning capacity—at multiple timepoints matching routine follow-up after TKA and THA. In addition, there is a lack of clarity whether gait can be assumed to be ‘normal’ one year after joint replacement (Naili et al., 2017; Bahl et al., 2018; Milner, 2009). Finally, little is known about how gait recovery compares to self-reported recovery of physical function (e.g., PROMs). Therefore, the aims of this study were threefold: (1) to investigate gait recovery at two and fifteen months after TKA and THA using inertial sensors, (2) to compare gait 15 months after TKA and THA with data from healthy participants, and (3) to compare recovery trajectories between objective gait parameters and self-reported scores physical functioning.

Materials & Methods

Participants

Individuals with end-stage OA scheduled for TKA (n = 24) or THA (n = 24) at the Sint Maartenskliniek participated in this study. A group of healthy controls (HC; n = 27) within the same age range of 50 to 75 years old was recruited from the community for reference purposes. Healthy participants had no pain in the lower extremities, nor were they familiar with a clinical diagnosis of knee or hip OA. All participants had to be able to walk for more than two minutes without the use of any assistive device. Exclusion criteria were: (1) joint replacement within a year following surgery (including revisions), or symptomatic OA in another weight-bearing joint than the joint scheduled for surgery, (2) BMI >40 kg/m2, and (3) any other musculoskeletal or neurological impairment interfering with gait or balance. Participants who received any other joint replacement to the lower extremities, or had a revision surgery within the period of fifteen months follow-up, were labeled as lost to follow-up. In these cases, data that had been collected until the time of the second surgery was still used for analysis. Written informed consent was obtained from all participants prior to testing. This study was exempt from ethical review by the CMO Arnhem/Nijmegen (2018-4452) as it was not subject to the Medical Research Involving Human Subjects Act (WMO). All study procedures were conducted in accordance with the Declaration of Helsinki.

Power calculation

Sample sizes were based on the smallest difference that we aimed to detect in this study, which was the difference in gait parameters between individuals 15 months after arthroplasty and HC. Effect sizes for this comparison were informed by studies from Senden et al. (2011) and Kluge et al. (2018). When using a standardized mean difference for stride length of 1.1, a power of 80%, and a significance level of 0.05, 22 participants were required per group. To account for potential drop-outs, 24 individuals were recruited for each study group.

Surgical procedure

TKA was performed using the medial parapatellar approach. All individuals scheduled for TKA received the Genesis II posterior stabilized knee prosthesis (Smith & Nephew, Memphis, TN, USA). The patella was resurfaced in 58% of the patients. THA was performed using the posterolateral approach. Specific types of hip implants differed among individuals scheduled for THA and are listed in File S1. In total, TKA was performed by seven different surgeons in this study, whereas THA was performed by ten different surgeons. All patients followed an enhanced recovery protocol with mobilization on the day of surgery and hospital discharge within two days.

All patients were referred to out-of-hospital physical therapy, which was focused on optimizing functionality, mobility, muscle power, coordination, stability, and walking improvement. Although physical therapy protocols were not standardized, patients usually continued physical therapy for 6–12 months, until their functional goals had been reached.

Demographic and clinical assessment

Severity of radiological OA was determined using Kellgren and Lawrence (KL) grades (Kellgren & Lawrence, 1957) as scored by JS and VB. Baseline anthropometric characteristics (e.g., body mass, height, and BMI) were obtained during the pre-operative screening visit. In addition, PROMs were assessed using the Knee Injury and Osteoarthritis Outcomes Score (KOOS) for TKA (de Groot et al., 2008) and Hip Disability Osteoarthritis Outcome Score (HOOS) (de Groot et al., 2007) for THA patients. More specifically, HOOS and KOOS subscales “Pain” and “Activities of Daily Living (ADL)” were used to represent pain and physical function. PROMs and gait were assessed pre-operatively –on the same day as the pre-operative screening visit –and at two and fifteen months follow-up. Follow-up measurements were initially set to take place at one year, but measurements were delayed with three months due to the COVID-19 pandemic. Timepoints of follow-up were chosen to match routine follow-up after TKA and THA in the Netherlands, and roughly reflect the moments when patients can walk independently without an assistive device (e.g., 2 months) and when full recovery has been achieved (e.g., 1 year). For HC, gait was investigated at only one occasion.

Gait protocol

Experimental procedures of the gait assessments were similar to the methods described in (Boekesteijn et al., 2021). Four inertial sensors (Opal V2, APDM Inc., Portland, OR) were attached to the dorsum of both feet, the waist (sacrolumbar level), and the sternum. Participants walked back and forth along a six meter trajectory making 180°  turns for a total duration of 2 min (Fig. 1). Gait tests were performed at comfortable, self-selected speed.

Figure 1 Overview of the experimental set-up and outcome parameters.

Wearable inertial sensors were used to capture gait parameters during a 2 min walk test over a six meter walkway with 180 degree turns. The figure is adapted from Boekesteijn et al. (2021) .

Data analysis

Raw inertial data was processed using validated Mobility Lab v2 software (Morris et al., 2019). Turning steps were separated from straight walking based on the gyroscope data of the lumbar sensor (El-Gohary et al., 2014). Gait parameters were calculated for each stride during steady-state walking phases, excluding the two steps preceding and following a turn. Parameters were summarized as mean value of all valid strides or turns. Based on non-redundancy and size of the difference between individuals with end-stage knee and hip OA and HC as found previously (Boekesteijn et al., 2021), the following outcomes were extracted (Fig. 1): (1) gait speed, (2) stride length, (3) cadence (4), step time asymmetry, (5) stride time variability, (6) peak turning velocity, (7) lumbar sagittal range of motion, (8) lumbar coronal range of motion, and (9) trunk coronal range of motion. Parameters were only evaluated for the TKA or THA group in case they were previously found to be sensitive to mobility impairment in knee or hip OA (Boekesteijn et al., 2021). For this reason, step time asymmetry, lumbar sagittal range of motion, and lumbar coronal range of motion were not evaluated in the TKA group.

Statistical analysis

Recovery trajectories of gait parameters and KOOS/ HOOS scores were visualized on group level by the mean and 95% confidence intervals (CI). Linear mixed models with gait parameters and KOOS or HOOS scores as dependent variable, time as two independent dummy variables (e.g., T2 and T15), and subject ID as random effect factor were constructed to investigate the effect of time on gait and KOOS/HOOS scores for TKA and THA separately. Addition of random slopes was evaluated, but these were not included in the final model for reasons of parsimony, as this did not contribute to a better model fit. Gait parameters of TKA and THA groups were compared with HC at 15 month follow-up using an independent samples t-test or non-parametric Mann–Whitney U test in case data was not normally distributed. Inferences of statistical significance were based on p < 0.05. Since multiple outcome parameters were used for the same construct (e.g., gait) we controlled the family-wise error rate using the Hommel procedure (Hommel, 1988), by adjusting the p-values for the number of gait parameters involved in each comparison. To assess discrepancies between gait and self-reported scores of physical function, we compared trajectories between gait speed, which was found to be most sensitive to gait impairment in knee and hip OA (Boekesteijn et al., 2021), and KOOS/HOOS-ADL scores. Meaningful improvements were defined as a change in gait speed >0.10 m/s (Bohannon & Glenney, 2014) and a change in KOOS/HOOS ADL score >20 points (Lyman et al., 2018). Data were processed in Python 3.8.3 and statistical analyses were conducted in RStudio 3.6.1 using the lme4 package (version 1.1-26) (Bates et al., 2015).

Results

Participant characteristics

The study groups did not differ significantly in age, sex, height, or BMI (Table 1). Compared to HC, body mass was significantly higher in individuals scheduled for TKA and THA. All individuals scheduled for TKA or THA had moderate to severe OA (KL grades 3 or 4). In total we had missing data for eleven participants. Three participants had a complication within the study window. For details regarding missing data and complications, see File S2.

Table 1 Baseline characteristics.

	TKA (n = 24)	THA (n = 24)	HC (n = 27)	Main effect	Post-hoc analysis	
Age (y)	63 [61, 66]	64 [62, 67]	66 [63, 68]	F(2,72) = 0.81, p = 0.448		
Sex (M:F)	12:12	16:8	13:14	χ2 (2, N = 75) = 2.07, p = 0.355		
Height (m)	1.73 [1.69, 1.77]	1.75 [1.72, 1.79]	1.72 [1.68, 1.75]	F(2,72) = 0.98, p = 0.381		
Body mass (kg)	84.6 [78.6, 90.6]	86.0 [78.1, 94.0]	75.7 [71.5, 80.0]	F(2,72) = 3.66, p = 0.031	TKA vs. HC: t(49) = 2.527 p = 0.015 THA vs. HC: t(49) = 2.428 ; p = 0.019	
BMI (kg/m2)	28.2 [26.6, 29.9]	27.9 [25.6, 30.2]	25.7 [24.5, 26.8]	F(2,72) = 2.91, p = 0.060		
KL score (I:II:III:IV)	0:0:8:16	0:0:6:18	–			
Notes.

TKA total knee arthroplasty

THA total hip arthroplasty

HC healthy controls

BMI body mass index

KL Kellgren Lawrence

Data are presented as mean (95% CI).

Recovery of gait after arthroplasty

Two months after surgery, gait speed, stride length, and cadence were not significantly different from baseline, both after TKA and THA (Table 2; Figs. 2A–2C). Peak turning velocity improved with 19.1 deg/s (95% CI [6.9–31.5]) in the first two months after THA, but not after TKA (Table 2). There were no changes in step time asymmetry within the first two months after THA (Table 2), nor were there changes in stride time variability after TKA and THA at this timepoint (Table 2). As for kinematics of the trunk, trunk coronal RoM was slightly lower two months after TKA (mean diff: −1.0 deg, 95% CI [−1.6 to −0.3]) compared to pre-operatively, whereas lumbar sagittal RoM was lower two months after THA (mean diff: −1.9 deg, 95% CI [−3.0 to −0.8]) (Table 2).

Table 2 Effects of time on gait parameters in the TKA and THA group.

	TKA (n = 24)	THA (n = 24)	
		Baseline –2 months	2 months –15 months		Baseline –2 months	2 months –15 months	
Gait parameters	Baseline (intercept)	Mean difference (95% CI)	P value	P corr	Mean difference (95% CI)	P value	P corr	Baseline (intercept)	Mean difference (95% CI)	P value	P corr	Mean difference (95% CI)	P value	P corr	
Gait speed (m/s)	0.99	−0.04 (−0.10, 0.03)	0.272	0.569	0.22 (0.15, 0.29)	<0.001	<0.001	0.96	0.04 (−0.02, 0.10)	0.245	0.514	0.14 (0.06, 0.20)	<0.001	0.003	
Stride Length (m)	1.16	−0.002 (−0.05, 0.05)	0.924	0.924	0.14 (0.09, 0.19)	<0.001	<0.001	1.11	0.05 (0.002, 0.10)	0.049	0.306	0.07 (0.02, 0.13)	0.013	0.076	
Cadence (steps/min)	102.8	−3.8 (−6.8, −0.8)	0.016	0.081	10.1 (7.0, 13.2)	<0.001	<0.001	103.0	−1.1 (−3.9, 1.7)	0.453	0.514	6.9 (3.8, 10.1)	<0.001	<0.001	
Peak turning velocity (deg/s)	164.0	8.5 (−6.5, 23.7)	0.275	0.759	17.4 (1.7, 33.0)	0.035	0.105	171.2	19.1 (6.9, 31.5)	0.004	0.033	11.1 (−2.5, 24.9)	0.121	0.484	
Step time asymmetry (%)	–	–	–	–	–	–	–	4.3	−0.7 (−2.0, 0.6)	0.292	0.514	−1.0 (−2.4, 0.3)	0.155	0.553	
Stride time variability (%)	2.3	0.2 (−0.2, 0.6)	0.380	0.569	−0.3 (−0.8, 0.1)	0.117	0.234	2.5	−0.1 (−0.5, 0.3)	0.514	0.514	−0.2 (−0.7, 0.2)	0.277	0.831	
Lumbar sagittal RoM (deg)	–	–	–	–	–	–	–	8.1	−1.9 (−3.0, −0.8)	0.001	0.013	0.1 (−1.1, 1.3)	0.870	0.870	
Lumbar coronal RoM (deg)	–	–	–	–	–	–	–	5.2	0.4 (−0.3, 1.3)	0.255	0.514	1.4 (0.6, 2.1)	0.001	0.010	
Trunk coronal RoM (deg)	7.8	−1.0 (−1.6, −0.3)	0.009	0.049	0.1 (−0.6, 0.9)	0.710	0.710	8.1	−0.6 (−1.3, 0.1)	0.087	0.439	0.1 (−0.6, 0.8)	0.797	0.870	
Notes.

TKA total knee arthroplasty

THA total hip arthroplasty

RoM range of motion

P corr Hommel adjusted p-value

Data are presented as mean (95% CI).

Figure 2 Recovery trajectories of gait parameters and PROMs.

Dots with error bars represent group means with 95% CI, whereas grey areas display HC group means with 95% CI. Individual datapoints are represented as small dots. Please note that dashed lines indicate linear recovery trajectories, which may deviate from the actual situation. Note: TKA, total knee arthroplasty; THA, total hip arthroplasty; HC, healthy controls.

Between two and fifteen months, large improvements in gait speed, cadence, and stride length were observed after both TKA and THA (Table 2; Figs. 2A–2C). For gait speed, the gain between two and fifteen months was 0.22 m/s (95% CI [0.15–0.29]) after TKA and 0.14 m/s (95% CI [0.06–0.20]) after THA. Peak turning velocity did not change significantly (mean diff: 17.4 deg/s, 95% CI [1.7–33.0], Pcorr = 0.105) between two and fifteen months after TKA. There were no significant improvements in turning velocity between two and fifteen months after THA (Table 2). Step time asymmetry did not change between two and fifteen months after THA. There were no changes in stride time variability, or trunk coronal RoM between two and fifteen months after TKA and THA (Table 2). Individuals after THA showed an increase of 1.4 degrees (95% CI [0.6–2.1]) in lumbar coronal RoM between two and fifteen months. Finally, none of the gait parameters were significantly different from HC at fifteen months after TKA and THA (Table 3; Figs. 2A–2I).

Table 3 Post-operative situation compared to HC.

			TKA vs HC		THA vs HC	
Gait parameters	HC (n = 27)	TKA –15 mo (n = 21)	Mean diff (95%CI)	Test statistic (df= 46)	P TKA−HC	P corr	THA –15 mo (n = 18)	Mean diff (95%CI)	Test statistic (df= 43)	P THA−HC	P corr	
Gait speed (m/s)	1.24 (1.18, 1.31)	1.19 (1.13, 1.24)	−0.06 (−0.15, 0.03)	1.31	0.197	0.569	1.16 (1.10, 1.22)	−0.08 (−0.18, 0.02)	1.70	0.096	0.441	
Stride Length (m)	1.32 (1.26, 1.37)	1.30 (1.25, 1.36)	−0.02 (−0.10, 0.07)	0.40	0.691	0.691	1.26 (1.20, 1.31)	−0.06 (−0.15, 0.03)	−1.32	0.194	0.581	
Cadence (steps/min)	113 (110, 117)	109 (105, 112)	−4 (−9, 1)	1.66	0.104	0.402	110 (107, 113)	−3 (−8, 2)	−1.13	0.265	0.794	
Peak turning velocity (deg/s)	205 (190, 220)	190 (174, 206)	−15 (−38, 8)	1.30	0.201	0.569	207 (190, 223)	1 (−23, 26)	1.40	0.909	0.909	
Step time asymmetry (%)	2.7 (1.9, 3.4)	–	–	–	–	–	2.6 (1.9, 3.2)	−0.2 (−1.3, 1.0)	−0.28	0.778	0.909	
Stride time variability (%)	1.8 (1.6, 2.0)	2.1 (1.8, 2.4)	0.3 (−0.1, 0.7)	1.68	0.099	0.398	2.1 (1.8, 2.4)	0.3 (−0.03, 0.7)	1.86	0.069	0.397	
Lumbar sagittal RoM (deg)	5.3 (4.7, 5.9)	–	–	–	–		6.3 (5.7, 6.8)	0.9 (−0.2, 2.1)	1.63	0.111	0.444	
Lumbar coronal RoM (deg)	8.1 (6.3, 9.0)	–	–	–	–		6.5 (5.3, 7.3)	-1.6 (−2.7, 0.1)	U = 165.5	0.074	0.397	
Trunk coronal RoM (deg)	6.6 (5.9, 7.3)	7.1 (6.2, 8.1)	0.5 (−0.7, 1.7)	0.89	0.379	0.691	7.5 (6.6, 8.5)	0.8 (−0.6, 2.1)	1.16	0.254	0.761	
Notes.

HC healthy control

TKA total knee arthroplasty

THA total hip arthroplasty

RoM range of motion

P corr Hommel adjusted p-value

Non-normal distributed data are presented in italic and are summarized as median (IQR) with median difference (95% CI). Test statistics represent either the t-value (normal data) or U (non-normal data).

Changes on PROMs after arthroplasty

Two months after TKA, individuals improved on all KOOS subscales, except for ‘Symptoms’ (Table 4). For all other subscales, self-reported scores showed large improvements (>20 points) with some individuals already reaching (sub)maximal scores (≥90 points) within the first two months (Fig. 2J & 2K). Further improvements were found for all KOOS subscales from two to fifteen months follow-up (Table 4). As for the HOOS, all subscales improved from baseline to two months after THA, as well as from two to fifteen months follow-up, with the largest magnitude of effects taking place in the first two months (Table 4).

Table 4 Patient-reported outcome scores for both groups at each timepoint.

		TKA (n = 24)		THA (n = 24)	
		Pre-operative –2 months	2 months –15 months		Pre-operative –2 months	2 months –15 months	
PROM scores	Baseline (estimate)	Mean difference (95% CI)	P-value	Mean difference (95%CI)	P-value	Baseline (estimate)	Mean difference (95% CI)	P-value	Mean difference (95%CI)	P-value	
HOOS/KOOS											
1) Symptoms	50	5 (−3, 14)	0.210	27 (18, 36)	<0.001	41	37 (30, 44)	<0.001	12 (4, 20)	0.007	
2) Pain	42	28 (20, 36)	<0.001	19 (10, 28)	<0.001	39	45 (39, 52)	<0.001	9 (2, 16)	0.017	
3) ADL	52	21 (13, 29)	<0.001	18 (10, 26)	<0.001	39	42 (35, 45)	<0.001	12 (4, 20)	0.004	
4) Sports/Recreation	16	22 (9, 34)	0.001	30 (17, 43)	<0.001	15	48 (39, 57)	<0.001	13 (3, 22)	0.014	
5) Quality of life	26	25 (17, 33)	<0.001	29 (20, 37)	<0.001	24	42 (32, 51)	<0.001	19 (9, 29)	<0.001	
Notes.

OA osteoarthritis

TKA total knee arthroplasty

THA total hip arthroplasty

HC healthy controls

ADL activities of daily living

Relation between recovery trajectories of gait parameters and PROMs

When comparing recovery trajectories of self-reported scores with gait parameters, substantial differences were observed (Fig. 2). Where KOOS and HOOS scores showed large improvements over almost all subscales in the first two months after surgery (Table 4), gait parameters generally improved between 2 and 15 months, with the exception of trunk-related gait parameters. More specifically, discrepancies between HOOS/KOOS-ADL scores and spatiotemporal parameters were present at two months after surgery. For gait speed specifically, there were no significant changes between baseline and two months after TKA and THA, while HOOS/KOOS-ADL improved with 42 points and 21 points, respectively. To illustrate, two months after surgery, 10/23 individuals after TKA reported meaningful improvements in ADL scores, while merely 4/23 showed a meaningful improvement in gait speed. Similarly, after THA, 20/23 individuals reported meaningful improvements in ADL scores at 2 months, with 10/23 individuals showing meaningful improvements in gait speed.

Discussion

This study evaluated the use of inertial sensors to monitor functional recovery after TKA and THA. In concordance with our previous work, that sensor-derived gait parameters show sensitive to knee and hip OA (Boekesteijn et al., 2021), this study showed that these parameters were also responsive to TKA and THA at two and fifteen months after surgery, and recovered to the same level as HC fifteen months after surgery. In addition, discrepancies between recovery trajectories of spatiotemporal gait parameters and HOOS/KOOS scores were observed, particularly at two months post-operatively.

Recovery trajectory of gait after TKA and THA

There were limited improvements in spatiotemporal gait parameters two months after TKA and THA, which is in agreement with previous studies (Senden et al., 2011; Bahl et al., 2018). However, the observed faster turning in absence of higher gait speed two months after THA is interesting, and may suggest that turning is more sensitive to short-term improvements in physical function after THA than gait speed. In contrast to these basic spatiotemporal parameters, normalization of trunk movement was found already two months after TKA and THA. Pre-operatively, individuals with knee OA may increase lateral trunk lean as a strategy to reduce knee joint loading and/or pain (Mündermann, Dyrby & Andriacchi, 2005; Hunt et al., 2008; Linley et al., 2010), which is no longer required two months after TKA. Increased lumbar RoM in the sagittal plane, in its turn, may serve as pre-operative compensation for individuals with hip OA to overcome pain and hip joint stiffness (Hurwitz et al., 1997; Lenaerts et al., 2009). Taken together, these results suggest that while two months is too early for meaningful recovery of spatiotemporal gait parameters, pre-operative compensations of the trunk and pelvis already disappear within the first two months after TKA and THA.

Large and clinically relevant improvements were observed on spatiotemporal parameters between two and fifteen months after TKA and THA. This is in agreement with literature investigating gait with inertial sensors one year after TKA (Fransen et al., 2019; Bolink, Grimm & Heyligers, 2015; Kluge et al., 2018) and THA (Bolink et al., 2016; Wada et al., 2019). Recovery of muscle strength (e.g., quadriceps and hip abductors)—which coincides with this period (Mizner, Petterson & Snyder-Mackler, 2005; Ismailidis et al., 2021) –may underly these improvements in walking capacity. As for trunk kinematics, both individuals after TKA and THA showed an increase in lumbar coronal RoM from two to fifteen months after surgery, which may relate to the restored ability of the hip abductors to control frontal plane pelvic movement (Bolink et al., 2016; Reininga et al., 2012). Compensations like lateral trunk lean, which limit pelvic RoM, are then longer required (Bolink et al., 2015). When combining these results with those of gait recovery at two months, it can thus be concluded that a wide range of sensor-derived gait metrics is responsive to TKA and THA, with spatiotemporal parameters and trunk kinematics each showing a distinctive recovery trajectory.

None of the gait parameters were different from HC mean values at fifteen months after TKA and THA. This in contrast with some earlier studies reporting remaining gait differences between HC and individuals one year after TKA (Kluge et al., 2018; Naili et al., 2017; Outerleys et al., 2021) or THA (Bahl et al., 2018). Although one year after arthroplasty is generally considered as endpoint of recovery, these differences between studies might be attributed to the longer follow-up time in our study. This seems like a reasonable explanation given that improvements in gait were larger in our study compared to these earlier studies (Kluge et al., 2018; Naili et al., 2017; Bahl et al., 2018). Our findings underscore the success of TKA and THA in improving physical functioning, and indicate that normal spatiotemporal gait parameters and normal trunk kinematics may be achieved 15 months after TKA and THA. Whether other aspects of gait, including lower-extremity kinematics and kinetics, also recover to the level of healthy controls remains to be elucidated. Despite our findings of full recovery after TKA and THA, current literature suggest that more advanced parameters, including lower extremity kinematics and kinetics, may still reveal deficits in gait one year after surgery (Naili et al., 2017; Bahl et al., 2018; Outerleys et al., 2021).

Relationship between PROMs and objective gait measures

Objective gait parameters showed a different recovery trajectory than subjective reports of physical function and pain. Scores on the KOOS and HOOS greatly improved within the first two months, while spatiotemporal gait parameters mainly improved between two and fifteen months after surgery. Similar discrepancies between PROMs, gait, and performance-based tests have previously been recognized in the literature (Bolink et al., 2016; Stevens-Lapsley, Schenkman & Dayton, 2011; Naili et al., 2017; Dayton et al., 2016; Luna et al., 2017; Mizner et al., 2011). For example, inverse recovery trajectories (i.e., early improvements in PROMs compared to worsening of performance-based outcomes) have been observed between KOOS/HOOS ADL scores and performance-based outcomes, including the 6 min walk test, stair climbing test, and timed up and go test, during the first month of recovery after TKA and THA (Stevens-Lapsley, Schenkman & Dayton, 2011; Dayton et al., 2016; Luna et al., 2017; Mizner et al., 2011). For sensor-derived gait parameters specifically, poor agreement with PROM scores has been found after TKA and THA (Bolink et al., 2016; Bolink, Grimm & Heyligers, 2015). On a similar note, Fransen et al. (2021) found that, although perceived walking ability and self-reported physical function improved, there were no improvements in quality or quantity of daily life gait three months after surgery. The current study adds that the discordance between gait parameters and self-reported function scores is most prominent at two months after surgery, with the exception of parameters related to trunk motion. The general consensus is that physical function subscales of PROMs assess a different domain than performance-based tests and gait analysis (Fransen et al., 2019). This discrepancy may first be related to a strong relation of physical function subscales with pain (Stevens-Lapsley, Schenkman & Dayton, 2011), as was also apparent from the similarity between the recovery trajectories of HOOS/KOOS Pain and ADL subscales in our study. One potential explanation for this is that improvements in pain directly translate to a more positive reflection on daily life performance, and that patients considered pain as the main limiting factor in their daily life activities. Second, these self-reported scores ask about experienced difficulty during a wide range of activities, rather than how they execute a specific activity, which is inherently different from what these gait parameters measure. Finally, there is evidence that objective parameters of physical function are more sensitive to remaining functional deficits after TKA than PROMs (Naili et al., 2017), which may attributed to early ceiling effects of PROMs. Since improving mobility—specifically walking—is an important goal of joint replacement (Lange et al., 2017), these sensor-derived parameters may thus add a relevant dimension to evaluation of physical functioning, although their clinical value still has to be demonstrated.

Limitations and future directions

This study has a number of limitations which merit attention. First, we measured gait recovery in a well-defined cohort of patients with unilateral osteoarthritis without pain complaints in any other joint or previous joint replacement. While this was relevant for the aims of the current study, this limits the generalizability of our findings. Second, in the present study, evaluation of physical function was limited to gait and turning in the present study while other daily life activities, including sit-to-stand transfers and stair climbing, are also relevant for physical functioning after TKA and THA (Dobson et al., 2013). Third, gait parameters in this study were limited to spatiotemporal parameters and gait-related trunk kinematics. Other parameters, such as knee and hip kinematics that can be derived from a different set-up of inertial sensors may provide additional information about gait recovery after TKA and THA, especially in light of remaining gait deficits (Bahl et al., 2018). While the current study touches upon the potential value of objective measurement of physical function, the actual value of clinical implementation of gait tests cannot be derived from our study results. Future studies with larger samples and a more diverse population are required to investigate the applicability of objective gait assessment systems to identify poor-responders. Another valuable direction would be to explore whether such data can be used to adjust patient expectations during clinical visits and to further tailor post-operative care. Finally, there is a need for studies employing inertial sensors for remote monitoring during daily life, which may not only enable more efficient (digital) healthcare pathways in the future, but may also contribute to data with greater ecological validity (Van Ancum et al., 2019; Takayanagi et al., 2019).

Conclusions

This study showed that objective gait measures derived from inertial sensors are responsive to TKA and THA. Not only speed-related parameters, but also turning and trunk motion provide important information about functional status before and at two and fifteen months after joint replacement. There were no remaining gait differences between individuals after TKA or THA and healthy participants at fifteen months. Recovery trajectories of objective gait data were different from those of KOOS and HOOS ADL subscales, with a marked discordance at two months after surgery. Altogether, these results strengthen the premise that sensor-derived gait metrics may provide meaningful information about recovery of physical functioning after TKA and THA that is not captured by self-reported ADL or pain scores.

Supplemental Information

Supplemental Information 1 Supplementary Information

Supplementary 1 - types of hip implants

Suplplementary 2 - missing data and complications

Click here for additional data file.

Additional Information and Declarations

Competing Interests

Author Contributions

Human Ethics

Data Availability

The authors declare there are no competing interests.

Ramon Boekesteijn conceived and designed the experiments, performed the experiments, analyzed the data, prepared figures and/or tables, authored or reviewed drafts of the article, and approved the final draft.

José Smolders conceived and designed the experiments, authored or reviewed drafts of the article, and approved the final draft.

Vincent Busch conceived and designed the experiments, authored or reviewed drafts of the article, and approved the final draft.

Noël Keijsers conceived and designed the experiments, authored or reviewed drafts of the article, and approved the final draft.

Alexander Geurts conceived and designed the experiments, authored or reviewed drafts of the article, and approved the final draft.

Katrijn Smulders conceived and designed the experiments, performed the experiments, analyzed the data, authored or reviewed drafts of the article, and approved the final draft.

The following information was supplied relating to ethical approvals (i.e., approving body and any reference numbers):

The CMO Arnhem/Nijmegen approved the study (2018-4452).

The following information was supplied regarding data availability:

The data and codes are available at Zenodo: Ramon Boekesteijn, José Smolders, Vincent Busch, Noël Keijsers, Alexander Geurts, & Katrijn Smulders. (2022). Data/Code: Objective monitoring of functional recovery after total knee and hip arthroplasty using sensor-derived gait measures (Version v2) [Data set]. Zenodo. https://doi.org/10.5281/zenodo.7014333.

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
