# Peer review of "Objective monitoring of functional recovery after total knee and hip arthroplasty using sensor-derived gait measures"

_PeerJ, doi:10.7717/peerj.14054_

## Round 0.1 · original submission · Major Revisions

While both reviewers believe that the data presented are relevant, they suggest important improvements regarding presentation and statistical analysis.

·

Basic reporting

In general, the article is clearly written with sufficient context provided. Clarification of the following points could improve understanding:
Line 101: exclusion criteria of ‘joint replacement within a year following surgery’ – does this mean you excluded people from analysis who had already been recruited and enrolled in your study? Is this a secondary joint replacement surgery (on the same/different joint)?
Line 147-148: You describe the steady-state walking phases as “excluding the steps preceding and following a turn” – how many steps before and after were excluded? How many steps were typically included in the turning phase?
Line 117: surgical procedure – You have a nice description of the TKA/THA procedure and supplementary info on implant type. Can you add information about how many different surgeons operated on the included participants?
Lines 182-184: describes different variable names than what are in the figure. It would help if these were consistent and/or if the text referred to the subplot letter within Figure 2 (e.g., Step time asymmetry (Figure 2E)).
Table 2,3,4 – using a dash to indicate negative and also to separate the lower and upper bounds of the confidence interval makes it difficult to read and see which CIs cross zero. You may want to consider using a comma to separate lower and upper bounds of confidence intervals instead of a dash.
Line 267-269: ‘normal gait can be achieved 15 months after TKA and THA’ – This statement is a bit strong given kinetic data showing post-surgery gait does not get to levels of healthy controls and thus perhaps is not ‘normal’ [Outerleys et al. Journal of Applied Biomechanics 2021, doi 10.1123/jab.2020-0051]

Experimental design

The research question – understanding the recovery trajectory of gait at multiple timepoints after TKA/THA, and comparison to the recovery trajectory of PROMS – is well laid out based on prior literature. However, it seems the analysis and results do not always match this research question. In lines 182-184 you describe baseline differences in gait parameters between the TKA/THA and HC groups, which do not help with understanding whether these variables can be used to monitor recovery. You do not use this information to select which variables to monitor at 2- and 15-month – does it make sense to examine changes in step time asymmetry, for example, if it is not different at baseline? It is also unclear why the gait speed and HOOS/KOOS ADL scores, but not other variables, were selected for the comparison between gait and PROMs. The comparison between trajectories of change in objective gait variables and subjective PROMs is interesting and fills the stated gap in the literature but the selection of these specific variables could be better justified.
My other main concern with the analysis is the large number of comparisons tested. While you have a nice power calculation section, if I am reading correctly, you created a separate linear mixed model for each gait variable and each PROM, and thus make a large number of comparisons without any adjustment for multiple comparisons. If I have misunderstood, please clarify in the article. If not, please adjust your results accordingly, consider removing variables from the analysis that are not different at baseline, and/or justify your reasons for not adjusting for multiple comparisons.

Validity of the findings

no comment

Reviewer 2 ·

Basic reporting

The study presents an interesting topic, with interesting data and results. However, in the present form, I find the scientific quality of the introduction and discussion of a moderate level. It generally stays at a superficial level with too many broad statements. The paper should be more precise on concepts and definitions, with sound and more complete argumentation, and with research questions and answers (conclusions) in line with what has been analysed.

Experimental design

The research question can be formulated more precise in line with what has been measured, analysed and compared.

Validity of the findings

Results seem to be valid and presented at a sufficient academic level. Research questions and answers (conclusion) need to be formulated more precise in line iwith what has been measured, analysed and compared.

Additional comments

Abstract
• Line 27: I suggest replacing ‘gait’ by ‘gait parameters’ or ‘sensor-derived gait measures’.
• Line 28: again, replace the first mentioning of ‘gait’ by ‘gait parameters’
• Line 35: replace ‘trunk kinematics’ by ‘gait-related trunk kinematics’. Also replace ‘spatiotemporal parameters’ by ‘spatiotemporal gait parameters’.
• Line 37: in which direction was the observed ‘discordance’?
• Line 40: Conclusions seem not to match the research questions, actual suitability of the application of wearable sensors in the orthopaedic clinic was not investigated. ‘complementary’ is also vague, as PROMS and sensor data appear not to be in line (at two months), how can it be concluded that sensor data complement PROMs, or even the other way around.

Introduction
• Line 64: what are dynamic situations of physical functioning, also in relation to ‘daily-life performance’? Hence, lines 62-68 could be formulated more precise (in terminology and definitions of concept).
• Line 74-78: These sentences basically indicate that measurement instruments should be sensitive, responsive and valid. Reliable might be added to this list. Not clear why ‘that is otherwise not available’ is required to add to this list. In addition, arriving at this part in the introduction there seems to be a gap in reasoning from physical functioning to gait and turning parameters, why are these parameters (such as?) important in the assessment of physical functioning? Assessment of recovery of gait may only be a part of the assessment of the recovery in physical functioning and what do (gait-related?) PROMs assess in this?
• Lines 89-91: First, 2 and 15 months may not be regular follow-up times in all (international) hospitals, so why would these time points necessary to investigate the responsiveness of gait parameters? Second, the research question is broadly formulated concerning the assessment of functional recovery. In line with my comments stated above, also the research question deserves to be formulated more precise as to the concepts and definitions of physical functioning, gait, gait parameters, (gait-related?) PROMs. What is measured and compared in the study?

Methods
• Participants: were participants or groups somehow matched on important confounding variables?
• Lines 104-107: was the study protocol ethically reviewed (in addition to the decision that the study was not subject to the Medical Research Involving Human Subjects Act)?
• Power calculation: I don’t understand the rationale of the power calculation. The introduction seems to point at responsiveness on the one hand (which would require different statistics and power calculations). On the other hand, the power calculations seem to be aimed at detecting differences between groups (in which specific gait parameters? Isn’t that relevant?) at 15 months. I could argue that the three groups are not different in their gait parameters at 15 months because the recovery period has been (successfully?) finished at 15 months, resulting in no differences between groups at 15 months (so why test for differences?). In addition, the research question seems to point at comparing recovery trajectories assessed by inertial sensors to those assessed by PROMs, which is not reflected in the sample size calculation. Both research question and power calculation need to be formulated and performed in line with each other.
• Surgical procedure: could more information be provided about the rehabilitation protocol and instructions for patients from the day of surgery to 15 months after surgery? Do protocols differ between THA and TKA?
• Statistical analysis: considering the statistical analyses, research questions seem to be: - do gait parameters and PROMs change over time (from pre-surgery to 2 and 15 months post surgery) in THA and TKA patients; - do gait parameters differ between THA / TKA groups and HC at baseline and 15 months; - is change in gait speed (why only gait speed?) from baseline to 2/15 months post surgery associated with the change in HOOS/KOOS scores (why look at meaningful improvements? What is deducted from the scatterplots?). The aim in the introduction should reflect these questions/analyses and should be substantiated in the introduction.
• line 156: what is a recovery pattern?
• Line 158: how was the dummy variable for time defined (only one?)? Was subject only included as random intercept or was also the random slope investigated for additional value for the model?
• Line 161: write down that ‘differences were considered significantly different from zero’, not data.

Results:
• Overall, results are clearly presented but could be improved by mentioning more of the actual data in addition to the indication of being significant or not.
• Line 176: replace ‘weight’ by ‘body mass’ (for the entire paper).
• Figures 2 and 3: I don’t find them very clear. Lines suggest linearity, which is not necessarily true. I suggest using bar charts with means and standard deviations (error bars). Furthermore, the figures suggest that 95%CI do not change over time? How are these calculated? (and they seem small in relation to the individual variation). Again, SD’s for each time point per group (or 95%CI per time point) seem more in line with the presentation of the actual data. Shouldn’t figures 2 and 3 be taken together (figure 2 does not include PROMs)?
• Figure 4: Presenting similar starting and end points for gait speed and PROM score is really suggestive, I suggest not to present it like this.

Discussion:
• Lines 278-279: I don’t agree with the statement that results may be explained by an overestimation of physical functioning compared to the actual performance. Before this it should be clear that both sensors and PROMs assess the same concepts, which may be more importantly not the case. Misjudgment in perceived versus actual ability is more complicated and context and task specific (see studies of Kluft and Weijer). Discussion on what the sensors and PROMs actually (tend to) measure and whether they can be compared or are additional to each other stays very superficial. Why are sensor derived data additional to PROMs in clinical practice (then it should be really clear what both methods measure)? What are the practical/clinical consequences?
• Line 287: I can’t agree with the statement that the evidence of the present study supports objective measurement of physical functioning. First, it is questionable whether both PROMs and the sensors measure ‘physical functioning’. Second, finding differences in time and between groups does not necessarily indicate that these objective measurements are supported. Personally, I agree that objective assessment of gait quality parameters in the clinic do have additional value, but the arguments presented in the discussion are not convincing and are limited to superficial statements.
• Lines 287-296: in contrast to the heading of the paragraph, actual limitations of the present study are not presented.

---

## Round 0.2 · Minor Revisions

The reviewers are happy with the revisions made. One reviewer has suggested an additional minor revision which will not require further review but that I ask you to take care of to finalise this valuable contribution to the literature.

·

Basic reporting

no comment

Experimental design

no comment

Validity of the findings

no comment

Additional comments

All my prior comments have been addressed and I have no further comments. Nice work!

Reviewer 2 ·

Basic reporting

-

Experimental design

-

Validity of the findings

-

Additional comments

The authors have greatly improved the manuscript and have sufficiently responded to the comments. I have one small additional comment and that is to indicate the meaning of p-corr in the tables. Overall it has become a very nice and interesting paper.

---

## Round 0.3 · accepted · Accept

Thank you for making this final improvement and congratulations on this contribution to the literature.